# Impact of Pelvic Anatomical Changes Caused by Radical Prostatectomy

**DOI:** 10.3390/cancers14133050

**Published:** 2022-06-21

**Authors:** Yoshifumi Kadono, Takahiro Nohara, Shohei Kawaguchi, Hiroaki Iwamoto, Hiroshi Yaegashi, Kazuyoshi Shigehara, Kouji Izumi, Atsushi Mizokami

**Affiliations:** Department of Integrative Cancer Therapy and Urology, Kanazawa University Graduate School of Medical Science, 13-1 Takara-machi, Kanazawa 920-8640, Ishikawa, Japan; t_nohara704@yahoo.co.jp (T.N.); shohei_k2001@yahoo.co.jp (S.K.); hiroaki017@yahoo.co.jp (H.I.); hyae2002jp@yahoo.co.jp (H.Y.); kshigehara0415@yahoo.co.jp (K.S.); azuizu2003@yahoo.co.jp (K.I.); mizokami@staff.kanazawa-u.ac.jp (A.M.)

**Keywords:** anatomy, complications, mechanism, prostate cancer, radical prostatectomy

## Abstract

**Simple Summary:**

After radical prostatectomy, the pelvic anatomy is altered such that the postoperative structure differs from the preoperative one, resulting in a variety of complications. In this review, the complications and mechanisms of pelvic anatomical changes associated with radical prostatectomy, as well as countermeasures, are outlined. An analysis of the anatomical mechanisms that cause complications after radical prostatectomy using imaging and other modalities is in progress. In addition, many surgical techniques that ensure the prevention of postoperative complications have been reported, and their usefulness has been evaluated. The preservation of as much periprostatic tissue and periprostatic structures as possible may lead to favorable postoperative functions, as long as the cancer condition permits.

**Abstract:**

During radical prostatectomy, the prostate is removed along with the seminal vesicles, and the urinary tract is reconstructed by dropping the bladder onto the pelvic floor and suturing the bladder and urethra together. This process causes damage to the pelvic floor and postoperative complications due to the anatomical changes in the pelvic floor caused by the vesicourethral anastomosis. Urinary incontinence and erectile dysfunction are major complications that impair patients’ quality of life after radical prostatectomy. In addition, the shortening of the penis and the increased prevalence of inguinal hernia have been reported. Since these postoperative complications subsequently affect patients’ quality of life, their reduction is a matter of great interest, and procedural innovations such as nerve-sparing techniques, Retzius space preservation, and inguinal hernia prophylaxis have been developed. It is clear that nerve sparing is useful for preserving the erectile function, and nerve sparing, urethral length preservation, and Retzius sparing are useful for urinary continence. The evaluation of pre- and postoperative imaging to observe changes in pelvic anatomy is also beginning to clarify why these techniques are useful. Changes in pelvic anatomy after radical prostatectomy are inevitable and, therefore, postoperative complications cannot be completely eliminated; however, preserving as much of the tissue and structure around the prostate as possible, to the extent that prostate cancer control is not compromised, may help reduce the prevalence of postoperative complications.

## 1. Introduction

One of the standard treatments for localized prostate cancer is radical prostatectomy (RP), which is widely performed worldwide. In RP, the prostate is removed along with the seminal vesicles, and the urinary tract is reconstructed by dropping the bladder into the pelvic floor and suturing the bladder and urethra together. This process causes damage to the pelvic floor and postoperative complications due to the anatomical changes in the pelvic floor caused by the vesicourethral anastomosis. Urinary incontinence and erectile dysfunction (ED) are major complications that impair quality of life (QOL) after radical prostatectomy. In addition, shortening of penile length and increased frequency of inguinal hernia (IH) occurrence have been reported. Although RP was initially performed with transperineal approaches, after the reports of anatomical RP and nerve-sparing (NS) techniques by Walsh et al., RP with retropubic approaches through a median incision in the lower abdomen became popular worldwide [1,2]. Since then, laparoscopic techniques have spread following their development and improvement [3], and robotic-assisted surgery is now widely performed [4]. In addition to the anterior approach, in which the anterior bladder is expanded and the operative field is secured [1], a Retzius-sparing posterior approach, in which the Retzius space is not opened and all operations are performed from the Douglas fossa, is now used for prostatectomy [5]. In this article, the effects of RP on pelvic anatomy and the complications and countermeasures resulting from these changes, including the mechanisms involved, are reviewed.

## 2. Penile Length after Radical Prostatectomy

### 2.1. Penile Length Measurement

The penis is an elastic organ, and its length varies depending on the conditions under which it is measured. Although it is highly desirable to measure the penis in an erect state, it is not easy to measure the penis in an erect state in a general outpatient clinic. Therefore, since the stretched penile length (SPL), which is measured with the penis fully extended, approximates the penile length at erection [6], many reports measure the SPL, which is relatively easy to measure [7,8,9,10,11,12,13]. To ensure uniform measurements, the temperature and body position at the time of measurement should be constant, and care should be taken to ensure that there are no changes in the thickness of the abdominal wall above the pubic bone when measurements are taken chronologically [13,14].

### 2.2. Chronological Changes of Penile Length after Radical Prostatectomy

The results of penile length shortening after RP are similar in all studies, although one report found that shortening continued up to 1 year postoperatively [9], whereas other reports claim that after shortening, penile length is regained over time [10,11,13]. Some of these studies report improvements to preoperative levels in approximately 6 to 12 months [11,13], whereas another study reported improvement in 3 to 5 years [10].

### 2.3. Penile Length Changes and Sexual Function

Factors affecting penile length changes after RP have been reported. It has been reported that changes in penile length correlated with sexual function as assessed by the five-item version of the International Index of Erectile Function (IIEF-5) [11]. There are also some reports that phosphodiesterase 5 inhibitors (PDE5i) administration could prevent shortening of penile length [12,15,16,17]. NS techniques reportedly had no effect on penile shortening [8,13,17]. The RP and control groups were asked about their awareness of penile shortening, QOL, and self-esteem. The results reported that awareness of penile shortening was higher in the RP group (55% vs. 26%), and that age, degree of ED, lower QOL, and lower self-esteem were related to the awareness of penile shortening [18]. Rather than penile shortening directly affecting sexual function, psychological influences, including self-esteem may be involved.

### 2.4. Mechanism of Penile Length Change after Radical Prostatectomy

The following mechanisms have been proposed to explain the changes in penile length after RP, based on magnetic resonance imaging (MRI) findings before and after surgery and at 1 year [13]. Anatomically, the Corpus spongiosum surrounding the urethra is an integral structure with the peripheral side continuous to the glans and the central side continuous to the bulb of penis (Figure 1A). At the time of vesicourethral anastomosis after prostatectomy, the bladder is dropped into the pelvic floor. Because the bladder is loosely pulled cephalad by the surrounding vascular pedicle and connective tissue, the anastomosis is pulled cephalad immediately after the vesicourethral anastomosis. As a result, the cavernous tissue, which is integrated with the urethra, is pulled in a pelvic direction, causing shortening of the penis (Figure 1B). Over time, the vascular pedicle and connective tissue that had been pulling on the bladder are gradually stretched, and after a period of approximately 1 year, penile length is expected to return to its original value, as the membranous urethra, which was being pulled cephalad, returns to its original position (Figure 1C). In animal experiments with rats, structural changes in the penis and sympathetic hypertonia, which may be caused by hypoxia resulting from damage to the penile cavernous nerve, have been observed and may be the cause of penile shortening [19]. Changes in penile length due to androgen have also been reported; there have been reports of shortening of the penis due to androgen deprivation therapy (ADT) [20,21], and improvement in penile length after the discontinuation of ADT, and a relationship between androgens and penile length have also been considered [22]. In the long term, it is possible that tissue changes within the penis under the influence of blood flow and hormones may affect penile length. It was also reported that an early postoperative intervention using a vacuum erection device reduced the shortening of penile length [23]. In the long term, there are many factors that affect penile length after RP, which may be one of the reasons for the variability in measurements between reports.

## 3. Erectile Dysfunction after Radical Prostatectomy

### 3.1. Pathophysiology and Anatomy of Erectile Function

Erection is caused by hyperemia in the penile corpus cavernosum, which is intricately controlled by nerves and blood flow. Blood flow to the penis is supplied primarily from branches of the internal pudendal artery (IPA), which are cavernous arteries and helicine arteries involved in erection. [24]. The first step in penile erection is tumescence that occurs following vasodilation of the arteries and simultaneous relaxation of the sinusoidal smooth muscle. The second step is veno-occlusion and rigidity which occur due to an increased pressure compressing the emissary veins against the tunica albuginea [24].

The nerves involved in erection are reported to form a mesh-like network around the prostate gland, forming a plate-like structure [25]. An MRI study in a pre-RP case reported that more than two-thirds of the nerve fibers are located in the posterolateral area, while the rest are located in the anterolateral and anterior position [26]. In addition, immunostaining of nerve fibers using RP specimens reported that 25% of all nerves were located in the anterolateral and anterior section of the prostate, suggesting that nerve damage is inevitable during RP, although it depends on the degree of NS. It is thought that nerve damage is inevitable during RP [27].

### 3.2. Surgical Technique including Nerve Sparing

Walsh et al. first reported an NS technique, assuming that the main cause of ED is damage to the nerve plexus around the prostate [28]. Robotic surgery with high-resolution endoscopes and articulated forceps has the advantage of allowing a detailed work in the narrow pelvis, and several meta-analyses have reported the advantage of preserving the erectile function over open or conventional laparoscopic RP (LRP) [29,30,31]. It has also been reported that robot-assisted RP (RARP) can provide appropriate functional preservation depending on the cancer status by using multiple levels of NS such as intrafascial, interfascial, and extrafascial approaches [32,33]. Tewari et al. described four levels of nerve preservation using the external prostatic venous plane as the index [34].There was a significant difference across different NS grades in terms of the percentages of patients who had intercourse and returned to baseline sexual function, with those that underwent NS grade 1 having the highest rates (90.9% and 81.7%) as compared to patients who received NS grades 2 (81.4% and 74.3%), 3 (73.5% and 66.1%), and 4 (62% and 54.5%) [34]. Similarly, a study by Patel et al. described five levels of nerve preservation using the landmark artery as the index [35]. Menon et al. reported good postoperative erectile function related to the 93% intercourse rate in men with no preoperative erectile dysfunction by removing the prostate with a line of dissection that extensively spares the perineural nerve plate around the prostate, known as the veil of Aphrodite [36]. Retzius-sparing RARP (RS-RARP) developed by Galfano et al. showed no difference in postoperative erectile function, although good postoperative urinary continence was verified [5,37]. Several meta-analyses also reported that postoperative erectile function was equivalent between conventional and RS-RARP [38,39,40]. Blood flow to the penis is considered to be supplied by the IPA and the accessory pudendal artery (APA) [41]. It was shown that the preservation of APA improves early ED in open RP in a study by Roger et al. [42]. Blood flow to the penis at pooled prevalence by meta-analysis was 61.9% IPA only, 5.4% APA only, and 32.8% from both; if APA is observed intraoperatively, it is preferable to preserve it [43].

## 4. Urinary Incontinence after Radical Prostatectomy

### 4.1. Pelvic Anatomy Affecting Urinary Incontinence after Radical Prostatectomy

The urethral sphincter is present distal to the prostatic apex and penetrates the pelvic floor muscles from the intra- and extra pelvic spaces, and the urethral sphincter itself is independent of the pelvic floor muscles [44]. The urethral sphincter has two layers: an inner layer composed of smooth muscle and an outer layer composed of striated muscle [45]. Outer rhabdomyoid fibers are omega-shaped and extend to the prostatic apex and anterior surface of the prostate [44,46,47]. The supporting structures of the male urethra can be divided into two main groups: anterior and posterior. The anterior urethral support structures include the pubourethral ligaments, comprising the pubovesical ligament, the puboprostatic ligament, and the tendinous arch of the pelvic fascia. These ligaments help stabilize the position of the bladder neck and external sphincter complex and anchor the membranous urethra to the pubic bone [48]. The posterior support consists of the perineal body (central perineal tendon), Denonvillier’s fascia, the rectourethralis muscle, and the levator ani complex [49,50]. The pelvic floor is formed by the levator ani muscle and its surrounding fascia, and although it is not directly connected to the urethra that penetrates it, it is thought to play an important role in urinary continence, especially under applied abdominal pressure, due to the urethral closure mechanism from outside the urethra [51].

### 4.2. Mechanism of Urinary Incontinence after Radical Prostatectomy

After RP, the supporting tissues near the urethra are damaged by the removal of the prostate. This damage is thought to increase the likelihood of stress urinary incontinence, primarily due to urethral sphincter dysfunction [52,53,54,55]. Bladder function has also been implicated in urinary incontinence after RP, with adverse effects due to detrusor overactivity and decreased bladder compliance [56]. When considering urinary continence mechanisms, it may be better to consider the resting situation and the situation under applied abdominal pressure separately. A urodynamic study before and after RP reported that the maximum urethral closure pressure at rest was reduced to about 40% of the preoperative pressure immediately after RP [57,58,59,60]. Subsequently, 1 year postoperatively, the maximum urethral closure pressure had improved, but not to preoperative levels, remaining at 80% of the preoperative level [58,59,60]. The following mechanisms have been reported to explain why urethral closure pressure decreases immediately after RP and improves over time (Figure 2) [60]. As reported for the mechanisms of penile shortening, the vesicourethral anastomosis causes the membranous urethra and urogenital diaphragm to be pulled cephalad after RP. To maintain maximum urethral closure pressure, the urogenital diaphragm, which compresses the urethra from the outside, must be in the proper position, which is probably usually the position before RP. After RP, the urethral closure pressure is expected to gradually recover as the vesicourethral anastomosis returns to its preoperative position over time [60]. During applied abdominal pressure, the pressure in the bladder increases, but at the same time the urethral closure mechanism causes an increase in the pressure in the urethra, which does not cause urinary incontinence. Urethroscopic observations revealed a shutter-like closure of the sphincter urethra in the anteroposterior direction of the body axis during abdominal pressure [46]. The mechanism of urethral closure during abdominal pressure has been observed in the pre-RP state, as well as in studies using transperineal ultrasound and dynamic MRI. The anterior wall of the rectum is reported to move in the direction of the pubic bone simultaneously with the applied abdominal pressure, and it is thought that this is observed mainly because the urethra is compressed in the anterior–posterior direction by the action of the levator ani muscle, resulting in increased pressure at urethral closure [61,62]. In a report examining pelvic anatomical changes during abdominal pressure after RP, the anterior wall of the rectum moved less toward the pubic bone than before RP, which is thought to affect the urethral sphincter closure pressure, resulting in an increased probability of stress urinary incontinence [62,63]. In fact, the abdominal leak point pressure test shows that urine leakage during abdominal pressure, which was rarely observed preoperatively, is observed in an increasing number of cases after RP, indicating that a decrease in urethral closure pressure during abdominal pressure occurs after RP [58,60]. MRI and transperineal ultrasound observations during abdominal pressure application have shown that the closure mechanism during abdominal pressure application does not work well in cases with high urinary incontinence after RP [62,64].

### 4.3. Maximum Urethral Preservation

A systematic review and meta-analysis reported that a preoperative long membranous urethral measured by MRI is favorable for urinary continence after RP [65]. It has also been reported that preserving the urethra as long as possible at the time of RP is advantageous for postoperative urinary continence [47,60,66]. The continence rates were 50.1% with full functional urethra preservation and 30.9% with the standard method 1 week after catheter removal (*p* < 0.0001) [47]. The two layers of muscle covering the urethral mucosa are thought to be associated with the generation of urethral closure pressure, and the longer the residual urethral length, the higher the urethral closure pressure, which may be advantageous for urinary continence [45].

### 4.4. Nerve-Sparing Procedure

Originally developed to preserve the erectile function, this NS technique has also been reported to provide excellent results in postoperative urinary continence [67,68]. Meta-analysis results show a consistent trend toward better urinary continence in patients undergoing NS up to 2 years postoperatively, with significantly less urinary incontinence reported with NS, especially in the early postoperative period up to 6 months postoperatively [69]. It has also been reported that the higher the grade of NS (dissecting the prostate in a line close to the prostate), the better the early postoperative urinary continence with a four-step NS technique. Return of continence ≤12 week postoperatively was achieved by 791 of 1417 men (55.8%); of those, 199 of 277 (71.8%) received NS grade 1, 440 of 805 (54.7%) NS grade 2, 132 of 289 (45.7%) NS grade 3, and 20 of 46 (43.5%) NS grade 4 [70]. There is some debate as to why postoperative urinary continence is better with NS, whether it is more important that the periprostatic nerves remain or that the periprostatic structures remain. During prostatectomy, the nerves around the prostate are damaged by the dissection maneuver, and it often takes time for the erectile function to return. It has been reported that the maximum urethral closure pressure was higher immediately after surgery in the case group receiving NS than in the case group without NS, suggesting that the preservation of as much peri-prostatic structure as possible may have a superior effect on urinary continence rather than the effect of the nerve itself [59].

### 4.5. Retzius Sparing Procedure

The Retzius-sparing procedure reported by Galfano et al. involves a transabdominal approach through the Douglas fossa and the removal of the prostate and vesicourethral anastomosis through this incision [5]. This technique leaves more periprostatic structures, including the anterior bladder lumen, than previously developed approaches, and better postoperative urinary continence has been reported compared to conventional methods [71,72,73]. Lee et al. reported that the continence recovery rate defined as the use of less than one safety liner per day in conventional and Retzius-sparing cases month 1 postoperatively was 9.0% and 45%, respectively, and by month 6 postoperatively, it was 77% and 98%, respectively [73]. In the Retzius-sparing procedure, the anterior wall of the bladder is not dissected, and after vesicourethral anastomosis, the anterior bladder wall is fixed in the same high position as before RP, without falling into the pelvis. This may be one of the reasons why the Retzius-sparing procedure is good for postoperative urinary continence, as it seems to work well for the urinary continence mechanism during abdominal pressure (Figure 3) [74]. In the conventional anterior prostatic approach, the vesicourethral anastomosis is thought to be pulled cephalad dorsally by the vascular pedicle of the bladder and the surrounding connective tissue. In the Retzius-sparing procedure, the anterior bladder wall is fixed in a higher position, which is thought to increase the force of traction of the vesicourethral anastomosis in the cephaloventral direction, which in turn increases the force to compress and drain the membranous urethra in the pubic bone direction and also maintains a high resting urethral closure pressure (Figure 4) [74]. Although transperineal RP does not open the anterior bladder space as the Retzius-sparing procedure, early urinary continence is reported to be inferior to that of conventional transabdominal RARP, possibly due to damage to the pelvic floor muscle complex that provides the pathway to the prostate gland [75]. Depending on the state of the cancer, a combination of techniques such as Retzius-sparing procedure, NS, and maximum urethral preservation may additively prevent the worsening of postoperative urinary continence [74].

**Figure 2 cancers-14-03050-f002:**
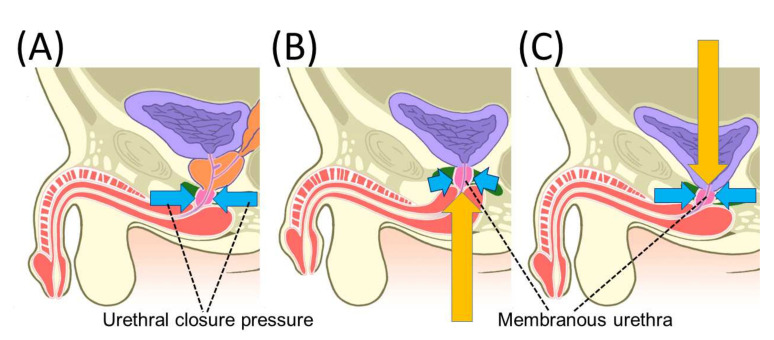
An illustration of chronological changes in pelvic anatomy after radical prostatectomy (RP). The membranous urethra is pushed proximally 10 days after RP and tends to be repositioned 12 months after RP. Yellow arrows indicate the direction of movement of the membranous urethra. The direction and length of the blue arrows image the direction and strength of the pressure on the membranous urethra. (**A**) Pre-operative; (**B**) 10 days after RP; and (**C**) 12 months after RP. Reprinted with permission from Ref. [60]. Copyright 2022 John Wiley & Sons, Inc.

### 4.6. Anterior and Posterior Reconstruction

Anatomically, the periurethral rabdosphincter is continuous dorsally to the median fibrous raphe and central tendon of the perineum, and further dorsally superiorly to the Denonvilliers’ fascia [76]. These structures are thought to form a dynamic lifting system that helps to contract the urethral sphincter, which is destroyed during prostatectomy. Rocco et al. reported that the reconstruction of these structures (posterior reconstruction) is effective in early urinary continence [46,77]. Tewari et al. reported that preservation of the puboprosthetic ligament and puboperineal muscle, re-fixation of the puboprosthetic ligament and vesicourethral anastomosis, and re-fixation of the bladder neck and tendinous arch (anterior reconstruction) are effective for early urinary continence [78]. Validation by randomized controlled trials (RCTs) has since been reported for these techniques. Two RCTs that tested the efficacy of posterior reconstruction in RARPs found no predominant improvement in the time taken for urinary continence recovery [79,80]. Another study reported that only the duration of improvement to 1 pad per day with posterior reconstruction was significantly shorter [81]. A RARP of an extraperitoneal approach to urinary continence reported that an RCT of an additional anterior suspension technique to the pubic periosteum alone was not effective for urinary continence [82]. In an RCT of open RP series of this anterior suspension combined with posterior reconstruction, the authors reported that urinary continence was better in the reconstruction group 1 and 3 months postoperatively, although there was no significant difference in urinary continence 6 months postoperatively [83]. An RCT comparing the combinations of anterior and posterior reconstructions with conventional techniques reported no advantage in urinary continence [84]. On the other hand, an RCT on the usefulness of the Advanced Reconstruction of the Vesico-Urethral Support (ARVUS) technique with strict posterior–anterior wall reconstruction reported better urinary continence than that achieved with conventional posterior wall reconstruction up to 1 year postoperatively when no pads were used to assess urinary continence [85]. Although there was no clear advantage of anterior and posterior reconstruction with respect to urinary continence, posterior wall reconstruction may be performed by many surgeons because the tension on the vesicourethral anastomosis is relieved by posterior reconstruction, which makes the vesicourethral anastomosis easier, and is expected to decrease the incidence of urinary bladder anastomosis failure [84].

## 5. Inguinal Hernia

### 5.1. Mechanisms and Risk Factors for the Development of Inguinal Hernia 

An increased incidence of IH has been reported after RP, with most cases occurring within 2–3 years after surgery [86]. The incidence of IH was reported to be significantly higher in the RP group (11.7% vs. 3.3%) in a meta-analysis between open and no treatment groups, and by surgical technique, with 13.7% for open RP, 7.5% for LRP, and 7.9% for RARP and a higher incidence after open RP [86]. Based on previous reports, the mechanism of IH after RP is speculated to be as follows: anatomic changes due to reattachment to the pelvic wall that occur after lower abdominal incision, opening of the Retzius space, and vesicourethral anastomosis associated with prostatectomy may contribute to the development of IH. In addition, the reported increased incidence of IH after pelvic lymph node dissection or cystectomy with lower abdominal incision suggests that external migration of the internal inguinal ring due to adhesions after lower abdominal incision may also be a factor in the development of IH [87,88]. Intra-abdominal observation during laparoscopic hernia repair shows scar contraction of the intra-abdominal wall at Hesselbach’s triangle and opening of the internal inguinal ring in the internal inguinal region [89]. In perineal RP and RS-RARP, the Retzius space is not opened, so the abdominal wall changes associated with reattachment do not occur. Therefore, IH due to the medial migration of tissue around the internal inguinal ring does not occur after vesicourethral anastomosis, and therefore the frequency of IH does not increase [90,91]. Risk factors for IH after RP have been reported to include older age, low body mass index (BMI), and high International Prostate Symptom Score (IPSS) [91,92,93].

### 5.2. Inguinal Hernia Prevention Techniques 

A method for preventing IH in open RP, which involves detaching the spermatic cord from the pelvic wall, detaching the vas deferens from the spermatic cord, and detaching and ligating the process vaginalis from the spermatic vessels, has been reported to significantly reduce the incidence of IH after RP [94]. In addition, the results of an RCT examining the efficacy of a trans-inguinal hernia prophylaxis procedure using non-absorbable sutures to suture the internal inguinal ring demonstrated the usefulness of this technique [95]. It has been reported that the same IH prevention method can prevent postoperative IH in RARP with an extraperitoneal approach, and it is thought that the method of preventing IH with an extraperitoneal approach is almost well established [96]. It has been reported that the occurrence of RARP by the transabdominal approach was prevented by opening the internal inguinal ring and filling it with hemostatic material in cases with process vaginalis remnants [97]. The following methods have been reported to prevent IH in conventional transabdominal RARP: (1) sufficient incision of the peritoneum around the internal inguinal ring; (2) separation of spermatic vessels; and (3) dissection of the vas deferens [98]. However, an RCT of a relatively similar prevention technique reported a lower but not significant incidence of IH [89]. In transabdominal approaches where the peritoneum does not cover the area around the inguinal ring postoperatively, prophylactic measures may have limited efficacy, and additional procedures such as repair of the open peritoneum or covering the inguinal ring with the peritoneum may be necessary.

## 6. Conclusions

After RP, pelvic anatomy differs from the preoperative one, resulting in a variety of complications. In this review, the complications and mechanisms of pelvic anatomical changes associated with RP, as well as countermeasures were outlined. ED and urinary incontinence are the main complications that impair QOL after RP. It is clear that NS is useful for preserving the erectile function, and NS, urethral length preservation, and Retzius sparing are useful for urinary continence. Preservation of as much periprostatic tissue and periprostatic structures as possible may lead to favorable postoperative outcomes and reduce postoperative complications, as long as the cancer condition permits. In addition to an accurate preoperative staging, detailed surgical planning and surgical techniques will be important.

## Figures and Tables

**Figure 1 cancers-14-03050-f001:**
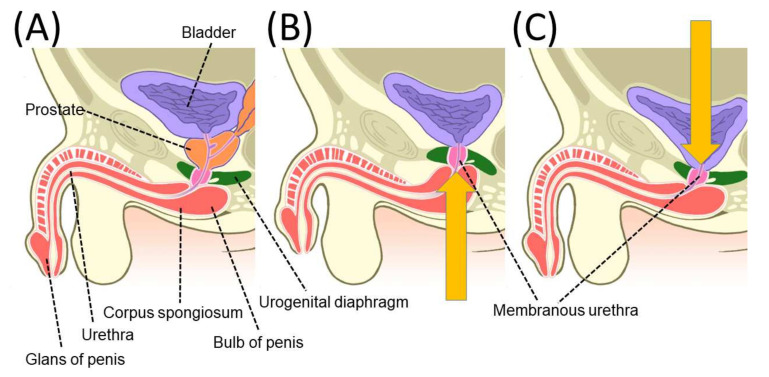
An illustration of chronological changes in pelvic anatomy after radical prostatectomy (RP). The membranous urethra is pushed proximally at 10 days after RP and tends to be repositioned 12 months after RP. Yellow arrows indicate the direction of movement of the membranous urethra. (**A**) Pre-operative; (**B**) 10 days after RP; and (**C**) 12 months after RP. Reprinted with permission from Ref. [13]. Copyright 2022 John Wiley & Sons, Inc.

**Figure 3 cancers-14-03050-f003:**
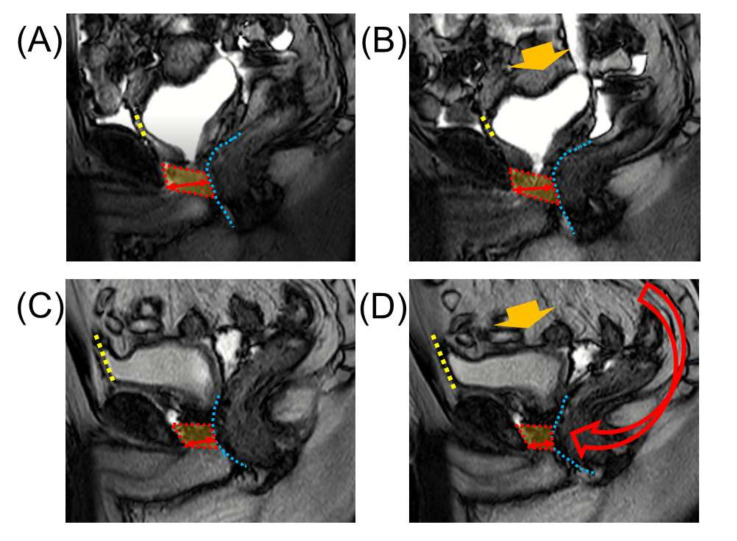
Dynamic mid-sagittal magnetic resonance imaging (MRI) after conventional robot-assisted radical prostatectomy (RARP): at rest (**A**) and with abdominal pressure (**B**). Dynamic mid-sagittal MRI after Retzius-sparing RARP: at rest (**C**) and with abdominal pressure (**D**). When applying abdominal pressure (orange arrow), the bladder is compressed caudally. At the same time, the pelvic organs are rotated forward (red arrow) with the anterior wall of the bladder (yellow dashed line) attached to the abdominal wall as a fulcrum, and the membranous urethra is compressed forward (blue dashed line). The thickness of the external urethral sphincter (two-headed red arrow) defined as the distance from the lowest point of the pubic bone to the anterior edge of the rectal wall (blue dashed line) was measured at rest and with abdominal pressure (orange arrow). The external urethral sphincter is indicated by the box surrounded by the red dashed line. Reprinted with permission from Ref. [74]. Copyright 2022 Springer Nature.

**Figure 4 cancers-14-03050-f004:**
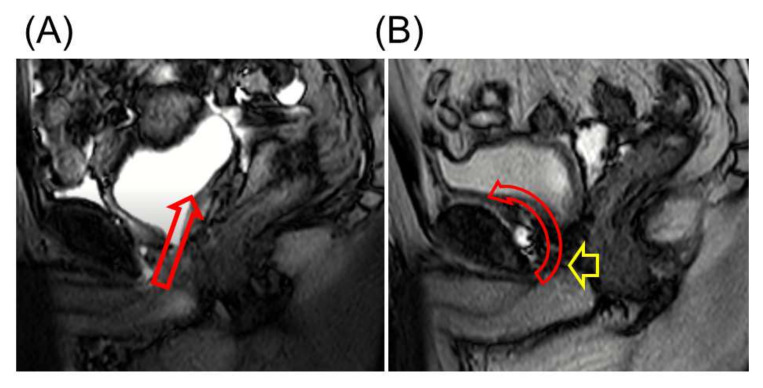
(**A**) Magnetic resonance imaging (MRI) after conventional robot-assisted radical prostatectomy (C-RARP). In C-RARP, the urethrovesical anastomosis is thought to be pulled cephalodorsally (red arrow) because the bladder vasculature is fixed from both dorsolateral sides. (**B**) MRI after Retzius-sparing RARP (RS-RARP). In RS-RARP, the anterior bladder wall is widely fixed, and the urethrovesical anastomosis is thought to be pulled cephalad ventrally (red arrow). After RS-RARP, the urethra is pushed in the direction of the pubic bone (yellow arrow), and the urethral closure pressure at rest may be higher than that after conventional RARP. Reprinted with permission from Ref. [74]. Copyright 2022 Springer Nature.

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
