# Peer review of "Impact of Pelvic Anatomical Changes Caused by Radical Prostatectomy"

_cancers, 2022, doi:10.3390/cancers14133050_

Round 1

Reviewer 1 Report

It has been a real pleasure to read this review, however I consider that the topic is not of sufficient interest to be published in a journal such as Cancers, I recommend to the authors its publication in a specific journal of genitourinary tumors.

Author Response

We appreciate the reviewer’s comments.

This review is a submission to the Special Issue "Prostate Cancer and Radical Prostatectomy; Controversies in Anatomy, Surgical Techniques and Outcome" organized by "Cancers". We believe that this paper is most appropriate for this project because it provides a comprehensive overview of the phenomena and complications that occur due to anatomical changes associated with radical prostatectomy, based on previous reports.

Reviewer 2 Report

The authors carried out a comprehensive review of the literature about the impact of radical prostatectomy on pelvic anatomy changes and the subsequent clinical aspects. The paper is well written, nevertheless some repetitions in the text have to be evaluated and modified in order to make the text easier to read (the word "reported" has been used so many times). Moreover I think that the authors must state that this is a review and if it is a systematic review the research methods.

Additional comments:

1.     What is the main question addressed by the research?

The impact of pelvic anatomical changes caused by radical prostatectomy
2. Do you consider the topic original or relevant in the field, and if
so, why?

The topic is relevant since different techniques have been developed in order to improve the functional outcomes after prostatectomy
3. What does it add to the subject area compared with other published
material?

This review summarizes in a comprehensive way the current available results
4. What specific improvements could the authors consider regarding the
methodology?

The strategy search has to be better explained
5. Are the conclusions consistent with the evidence and arguments
presented and do they address the main question posed?

Yes
6. Are the references appropriate?

Yes
7. Please include any additional comments on the tables and figures.

The figures attached are clear and they can hemp to better understand the anatomical changes after prostatectomy

Author Response

We appreciate the reviewer’s comments.

The paper has been reviewed again and revised for readability. Corrections have been made in red. At the end of Introduction, we mention that “In this article, the effects of RP on pelvic anatomy and the complications and countermeasures resulting from these changes, including the mechanisms involved were reviewed.”

Reviewer 3 Report

The review is interesting and addresses important clinical issue of post-prostatectomy complications. I have some suggestions which may make it more informative and useful for readers: (1) authors describe in detail mechanism of most common complications such as urinary incontinence and erectile dysfunction but do not provide the rates of these events after different types of prostatectomy (they provide the complication rate only for inguinal hernia). Providing the ranges of these common complications in the text or better in the form of the table would be valuable for readers. (2) Both the abstract and the conclusion section could be more informative rather then descriptive and repetitive (very close in content to the introduction section).

Author Response

We appreciate the reviewer’s comments.

Added specific numerical results from each technique according to the reviewer's comments in blue.

The following text was added in blue to the summary and conclusion as specific comments, “It is clear is that nerve sparing is useful for preserving erectile function, and nerve sparing, urethral length preservation, and Retzius-sparing are useful for urinary continence”.

Round 2

Reviewer 1 Report

It has been a privilege to review this article. These data will be of great value in clinical practice helping to reduce postoperative complications, properly educating the patient before undergoing a surgical procedure